# NuMY—A qPCR Assay Simultaneously Targeting Human Autosomal, Y-Chromosomal, and Mitochondrial DNA

**DOI:** 10.3390/genes14081645

**Published:** 2023-08-18

**Authors:** Catarina Xavier, Charlotte Sutter, Christina Amory, Harald Niederstätter, Walther Parson

**Affiliations:** 1Institute of Legal Medicine, Medical University of Innsbruck, 6020 Innsbruck, Austria; charlotte.sutter@irm.uzh.ch (C.S.); christina.amory@i-med.ac.at (C.A.); harald.niederstaetter@i-med.ac.at (H.N.); 2i3S—Institute for Research and Innovation in Health, University of Porto, 4099-002 Porto, Portugal; 3Forensic Science Program, The Pennsylvania State University, University Park, PA 16801, USA

**Keywords:** DNA quantification, human nuclear DNA, Y chromosome, mitochondrial DNA

## Abstract

The accurate quantification of DNA in forensic samples is of utmost importance. These samples are often present in limited amounts; therefore, it is indicated to use the appropriate analysis route with the optimum DNA amount (when possible). Also, DNA quantification can inform about the degradation stage and therefore support the decision on which downstream genotyping method to use. Consequently, DNA quantification aids in getting the best possible results from a forensic sample, considering both its DNA quantity and quality limitations. Here, we introduce NuMY, a new quantitative real-time PCR (qPCR) method for the parallel quantification of human nuclear (n) and mitochondrial (mt) DNA, assessing the male portion in mixtures of both sexes and testing for possible PCR inhibition. NuMY is based on previous work and follows the MIQE guidelines whenever applicable. Although quantification of nuclear (n)DNA by simultaneously analyzing autosomal and male-specific targets is available in commercial qPCR kits, tools that include the quantification of mtDNA are sparse. The quantification of mtDNA has proven relevant for samples with low nDNA content when conventional DNA fingerprinting techniques cannot be followed. Furthermore, the development and use of new massively parallel sequencing assays that combine multiple marker types, i.e., autosomal, Y-chromosomal, and mtDNA, can be optimized when precisely knowing the amount of each DNA component present in the input sample. For high-quality DNA extracts, NuMY provided nDNA results comparable to those of another quantification technique and has also proven to be a reliable tool for challenging, forensically relevant samples such as mixtures, inhibited, and naturally degraded samples.

## 1. Introduction

The development of assays for combined typing of multiple genetic marker systems for forensic applications has increased markedly in recent years. Taking advantage of the high throughput and multiplexing capabilities of massively parallel sequencing (MPS) techniques, forensic scientists can now simultaneously analyze autosomal, Y-chromosomal, and even mitochondrial (mt) DNA sequences. This enables, for example, the development of new assays combining identification, DNA phenotyping, and lineage prediction in a single reaction [1,2,3,4,5,6,7]. Due to the higher sensitivity of these new tools to input DNA amounts and particularly to balance primer concentrations, it becomes necessary to quantify the different targeted DNA components in the sample of interest. Furthermore, the simultaneous quantification of autosomal, Y-chromosomal, and mtDNA from a sample provides more accurate information on the quantity of DNA for all possible forensically relevant markers and thus better supports economical decision making on downstream applications whilst sparing laboratory and human resources.

Generally, assessing the DNA content of extracts from unknown samples by quantitative real-time PCR (qPCR) is a common procedure in forensic genetics laboratories to ensure that the most appropriate downstream application is taken for these samples in each case. Furthermore, accurate DNA quantification enables optimized DNA input for genotyping, saving precious extract and reducing or even avoiding artifacts in capillary electrophoretic STR typing and Sanger sequencing approaches as well as in MPS-based analyses. In addition to analyzing nuclear DNA (nDNA) markers, an increasing number of laboratories are becoming familiar with mtDNA sequencing and are recognizing its forensic value, especially for highly degraded samples. This is particularly supported by the provision of commercially available mtDNA kits for MPS.

Quantitative PCR tests currently available on the market and in the literature vary in the choice of targets and their level of multiplexing. To date, the most commonly used commercial kits only co-quantify autosomal and male-specific regions of the human genome [8,9]. These kits include two or three quantification targets, which can provide both quantitative data and information about nDNA degradation. The choice of commercial kits for the quantification of mtDNA [10,11,12] is limited, which motivated several groups to develop dedicated single- and multiplex qPCR tools [13,14,15,16,17,18,19,20,21,22,23]. However, to our knowledge, only one qPCR assay has so far covered human autosomal, Y-chromosomal, and mtDNA target sequences in a single reaction [15].

Here, we present NuMY, a novel multiplex qPCR assay developed for the parallel quantification of nDNA and mtDNA by simultaneously amplifying autosomal, Y-chromosomal, and mtDNA targets from human tissues. The Y-chromosomal target allows for the assessment of the male contribution to mixtures of both sexes. In addition, NuMY also includes an internal positive control (IPC) for inhibitor detection. NuMY is based on SD quants [22] and extends this assay by substituting one of its two mtDNA targets with YRS, a multi-copy region located on the Y-chromosome [24]. NuMY was evaluated using pre-quantified high-quality human DNA samples, and its results were juxtaposed with a non-qPCR-based quantification method. The assay’s practicality in routine scenarios was tested on mock casework extracts and challenging specimens like bone remains and hair samples. Mixtures of varying female-to-male DNA ratios were prepared to assess the detection of minor male components. Finally, the assay’s inhibitor tolerance was tested using hematin, humic acid, and indigo carmine to benchmark its performance for forensic samples.

## 2. Materials and Methods

### 2.1. Assay Design and qPCR Settings

NuMY is based on the previously developed qPCR assay SD quants [21]. The autosomal multi-copy target (nuRNU), one mitochondrial target (mtND1) [25], and the IPC were maintained and combined with YRS, a male-specific multi-copy target located on the human Y chromosome that was taken from the literature [24]. Contrary to the previously developed SD quants, which provided an evaluation of mtDNA degradation status, NuMY does not integrate two differently sized mtDNA targets; thus, this feature cannot be explored. However, the presence of two nuclear DNA targets (despite being located in different chromosomes) can give an indication of the nuclear DNA degradation status.

The feasibility of all primer and probe sets (Table 1) was first verified in singleplex qPCR. Subsequently, all primers and probes were combined into a multiplex reaction, and their concentrations were optimized (Table 1). The 20 µL qPCR assays comprised 2 µL DNA extract or standard, 10 µL 1X TaqMan Advanced Fast Master Mix (Thermo Fisher Scientific, Waltham, MA, USA; TFS), and 8 µL pre-made NuMY master mix (Table 1). Thermal cycling was performed in an ABI Prism 7500 instrument (TFS) with the following settings: initial template denaturation and enzyme activation at 95 °C for 20 s, followed by 40 cycles of denaturation at 95 °C for 3 s, and annealing/extension at 60 °C for 30 s. Fluorescence recording in the instrument’s FAM, VIC, NED, and ROX channels occurred during all annealing/extension steps. We used the 7500 Software (v2.0.6 or v2.3, both TFS) to acquire and analyze the raw data. Baselining was set to auto, and the following fluorescence thresholds were set for cycle-of-quantification (Cq) determination: 0.07862 for nuRNU; 0.02516 for YRS; 0.01239 for mtND1; and 0.03163 for the IPC. The raw fluorescence values were normalized to the ROX signal.

### 2.2. Samples

Different samples (all tested in duplicates) were used to assess NuMY’s performance with high-quality/quantity DNA, challenging DNA, inhibited samples, and mixtures. Here, we describe each set independently in the sections below.

#### 2.2.1. DNA Standards and Calibration Curves

As reference material for nuRNU and YRS quantification, a commercially available, pre-quantified total genomic (g)DNA (i.e., nDNA + mtDNA) preparation from multiple male donors was used (G1471; Promega, Madison, WI, USA). The mtDNA content of this standard was determined relative to a synthetic, double-stranded mtND1 target as described in [22]. Therefore, all three calibration curves were based on a single, eight-step, fivefold dilution series of the gDNA reference material. All standard curve dilutions were analyzed in triplicate and covered gDNA input ranges of 62.5 to 0.0008 ng/µL (125 to 0.0016 ng in the 20 µL reaction; nuRNU and YRS) and 972,568 to 12.5 mitochondrial genome equivalents (mtGE) per µL (1,945,136 to 24.9 mtGE per assay; mtND1). One mtGE corresponds to a double-stranded mtND1 target determined by NuMY.

#### 2.2.2. DNA Controls

To assess the overall performance of the assay, gDNA controls were analyzed and the NuMY results were compared to those of the non-qPCR-based Qubit dsDNA HS assay kit (TFS). Here, commercially available human gDNA controls (2800M, Promega; 007, TFS) with known DNA concentrations and Coriell samples from human gDNA collections covering different biogeographical ancestries were considered (NA07029, NA06994, and NA07000 of European descent—all CEPH; NA10540 from the Solomon Islands, Oceania; NA11200 from Peru, South America; and NA18498 from Nigeria, Africa). 

#### 2.2.3. Model Inhibitors

Aiming to test the sensitivity of the IPC toward PCR inhibitors, replicates of 1 ng of gDNA control 2800M were spiked with three well-known inhibitors: hematin, humic acid, and indigo carmine. An inhibitor concentration series was tested to gauge the assay’s sensitivity threshold. Hematin and humic acid were tested at assay concentrations ranging from 100–3.125 µM and ng/µL, respectively, and indigo carmine at a range spanning 3–0.09375 mM (Appendix A). 

#### 2.2.4. Challenging Samples

Challenging samples were considered in three categories: (1) mock casework samples, (2) hairs (roots and hair shafts), and (3) bone remains. Seven samples from six GEDNAP proficiency exercises were used as mock single-source casework samples (Table 2). More details on the biological origins of these samples can be found in Appendix A. Hair root samples were collected from five individuals (three men and two women), and two of them were also sampled for hair shafts (one of each sex). The bone remains used in this study dated from different time periods: (1) a 30-year-old bone from Austria that was used for teaching purposes; (2) one medieval bone sample (800–1500 years post mortem) coming also from Austria [26]; and (3) one 2000-year-old bone from Colombia [27]. These bone samples were extracted using two different methods: a) a DNA extraction protocol for bones that is widely applied within the forensic community [28,29] and b) a newer protocol that targets DNA segments of smaller size, which was developed within ancient DNA research [30]. More details on all challenging samples are provided in Appendix A.

#### 2.2.5. Mixtures

Defined mixtures of female and male source gDNA extracts were tested to gauge the detection of the male component on a predominantly female background via the analysis of the nuRNU and YRS quantification results. Male gDNA control 007 and a female sample with a higher DNA concentration were used at 1:1, 1:4, 1:9, and 1:19 ratios (Appendix A). The female sample was provided with voluntary informed consent under the VISAGE project (https://www.visage-h2020.eu accessed on 15 August 2023). 

### 2.3. Data Analysis

Basic data analysis was performed with Microsoft Excel, further statistical testing was performed in R [31], and graphs were created using routines implemented in base R or the R package ggplot2 [32]. Human specificity of the primer and probe sets was tested by using NCBI’s Primer-BLAST (https://bit.ly/40liayT accessed on 15 August 2023) by querying the ‘Refseq representative genomes’ database for the following taxonomical groups: great apes, mammals, birds, reptiles, amphibians, teleosts, cartilaginous fishes, insects, and arthropods. Furthermore, we mapped the primer and probe sequences (each with up to three mismatches) onto the GRCh38 genome assembly to assess the number of potential target sequences for each of the four NuMY targets. For this purpose, we used an R script that made use of the Bioconductor packages DECIPHER [33] BSgenome.Hsapiens.UCSC.hg38 [34], and dependencies thereof.

## 3. Results and Discussion

### 3.1. In Silico Characterization of the Quantification Modules

An error-tolerant search for the NuMY primer and probe binding sites in the GRCh38 genome assembly revealed differing results for YRS and nuRNU. The search for the YRS primer and probe binding sites yielded a large number of hits on the Y chromosome with expected amplicon sizes between 115 and 122 bp. We identified 44 perfectly matched primer and probe trios (all producing 117 bp amplicons), 209 target sequences showed a single mismatch in one of the primer or probe binding sites, 387 had a total of two mismatches, and 374 featured three mismatches (Appendix A). The remaining 670 hits on the Y chromosome had between four and nine mismatches. Another 473 possible YRS target sequences were found on other chromosomes, all of which had at least three mismatches. Given this large number of imperfect templates and the fact that both PCR success and probe binding highly depend on the number, location, and nature of the non-canonical base pairings, it is de facto impossible to determine the number of effective YRS targets even in a single human genome. Complicating matters further, the initial DNA quantity appears likely to be another influencing variable.

For nuRNU, the picture was different. All 13 perfect template sequences had a length of 70 bp and were located on chromosome 17. We were able to identify another 70 bp target on chromosome 10, which had a single, 5’-terminal G::T mismatch in the RNU forward primer, which should exert only a mild effect on primer binding. The remaining 13 hits were distributed across several chromosomes and had two to nine base mismatches, many of which are likely to affect oligonucleotide hybridization or primer extension (Appendix A). Overall, the nuRNU quantification module appears to be more stable in terms of the effective number of templates per genome when compared to YRS.

A series of Primer-BLAST analyses were performed to test the NuMY quantification modules for their human specificity. The synthetic IPC system yielded no predicted PCR products, mtND1 only for the intended single target sequence on the human mitochondrial genome, and YRS yielded results for the human and chimpanzee genomes. By contrast, this analysis showed hits for nuRNU in the genomes of different animal species. The (non-exhaustive) list of mammalian species with perfect 69–71 bp long nuRNU target sequences returned by our NCBI representative genomes database search comprised 26 non-human entries (Appendix A). When analyzing samples of uncertain human origin, nuRNU results should, therefore, be evaluated in the context of the results of the other two quantification targets.

### 3.2. Single- vs. Multiplex Assay Performance

Quantification standard curves were generated in the single- and quadruplex qPCR assay layout by plotting the measured Cq values against the decadic logarithm of the respective DNA input values (Appendix A). In no case did the multiplexing level show a statistically significant effect (ANOVA, *p* > 0.05) on the slopes of the linear regression lines and, consequently, also not on the average apparent PCR efficiency of a single cycle in the early phase of the amplification, which can be estimated as:E = 10^(−1/slope)^ (or as percentage Y = (E − 1)∙100).(1)

### 3.3. Working Range, PCR Efficiency, and Replicate Variation in Standard Curves

For all three quantification systems, dilution series covering more than five orders of magnitude were analyzed for linearity, as required by the MIQE guidelines for the working range of a qPCR assay. Seven calibration curves each were examined, and all standards were run in triplicate, resulting in 21 readings per standard concentration and target. All standard curves presented an R2 > 0.98 with mean values of 0.997 for both nuRNU and YRS and 0.999 for mtND1, which is in line with the MIQE guidelines (Appendix A) [35,36]. 

Quadruplex NuMY PCR efficiency values fell within the expected MIQE interval of 90–110% for all seven runs in all targets, with average values of 96.8%, 98.7%, and 106.2% for nuRNU, YRS, and mtND1, respectively. In addition, we calculated the differences between the Cq values of adjacent fivefold dilutions and compared them with the theoretical number of cycles (n) with 100% PCR efficiency (E = 2) that would be required to compensate for the applied dilution [28]. On average, these ΔCq values amounted to 2.22 (mtND1), 2.38 (nuRNU), and 2.35 (YRS) cycles. The ratio between these experimentally determined results and the theoretical value of n being log2(5) ≈ 2.32 was 0.96 (mtND1), 1.02 (nuRNU), and 1.01 (YRS). These results reinforce the log-linearity of all three multiplexed quantification systems over the entire template input range tested and underline the robustness and repeatability of the assay.

The Cq values showed some variability among replicates and runs. This particularly applied to mtND1 and nuRNU (Figure 1A), with mean standard deviations of 0.66 and 0.75, respectively. Contrarily, YRS yielded more consistent Cq values (mean standard deviation: 0.31). The exact reason for these differences among quantification systems is unknown. However, the results for mtND1 and nuRNU are only slightly above the ±0.5 Cq values stipulated by the MIQE guidelines.

### 3.4. Performance with High-Quality DNA Templates

The ratio between the nuRNU and YRS quantification result pairs provides a baseline to understand the variability of these two targets in samples of adequate gDNA quantity and quality. Under optimal conditions, a value of 1 is expected for male source samples. However, some variability is anticipated for biological samples. Therefore, we used two additional control DNA preparations and six Coriell samples to assess the overall performance of the assay in this regard. These samples all contained non-limiting amounts of gDNA, but beyond that, DNA fragmentation also appeared to be of relevance. The Coriell samples went through several freeze–thaw cycles, which may have had adverse effects on DNA stability.

The two nuclear DNA markers were amplified at different fragment sizes (nuRNU: 70 bp; YRS: 117 bp, Table 1), which resulted in distinct requirements for the minimum length of their template molecules. Therefore, an increased nuRNU/YRS ratio may be indicative of (substantial) nDNA degradation.

For most of the high-quality male source samples (Table 2), we found ratios between 0.5 and 2, but for the majority of them (4 out of 7) the ratio was closer to 2, a result well above the expected value of 1. The reason for this observation may be variations in the copy number of the multi-copy target sequences (nuRNU, YRS; Appendix A) [24,37] that are known to vary between individuals, particularly in the Y chromosome due to its excess in palindromic sequence stretches [38].

Results that vary from this range should be flagged for further investigations of that specific sample. In particular, sample NA18498 showed signs of template degradation due to a slightly increased nuRNU/YRS ratio (2.30), a finding that may be explained by repeated freeze–thaw cycles.

For comparison purposes, all Coriell samples were also quantified using the Qubit dsDNA HS Assay kit (TFS), resulting in small and statistically insignificant differences (Wilcoxon signed-rank test, *p* > 0.05, Figure 1B).

### 3.5. PCR Inhibitors

The ability to detect PCR inhibitors at low concentrations is becoming increasingly relevant for forensic applications, as MPS kits are less tolerant of inhibiting substances [7,39,40,41]. Three known PCR inhibitors were spiked into 2800M replicates of 1 ng, and the performance of NuMY was tested in terms of detection and tolerance to these inhibitors (Figure 1C). To establish a comparison interval for the expected IPC-Cq variation in neat samples, we analyzed all standards used in the seven qPCR runs (n = 179) and calculated an average IPC-Cq value and a probable interval surrounding it by ± 0.5 Cq values. We considered all inhibitor-spiked samples with IPC-Cq values outside this range to be flagged for potential PCR inhibition.

As shown in Figure 1C, the hematin series yielded well-reproducible IPC-Cq values within the expected mean ± 0.5 Cq interval, suggesting that the NuMY assay is tolerant to the presence of hematin at the tested concentrations. For the humic acid series, we observed two replicates outside the expected interval at inhibitor concentrations of 50 and 25 ng/µL. Finally, all indigo-spiked replicates either failed to amplify detectable amounts of product (i.e., full PCR inhibition) or showed IPC-Cq values below the comparison interval. This counter-intuitive effect is caused by an interference of the indigo dye with fluorescence detection, which simulates too high template DNA quantities [42].

When compared to other quantification assays such as the Quantifiler trio (TFS) or the Plexor HY system (Promega), the NuMY assay was less sensitive to hematin with no visible alteration of the IPC amplification with increasing hematin input (tested range: 100–3.125 µM in the assay). Quantifiler trio has been reported to detect the presence of hematin at ≥100 µM [8] and Plexor HY at ≥25 µM [9]. When considering humic acid, the NuMY assay was closer to the values found for Plexor HY (15 ng/µL), showing alterations to IPC amplification for one of the replicates at 50 and 25 ng/µL, respectively. However, since both NuMY 100 ng/µL replicates were not flagged for inhibitor presence, these results should be taken with caution and a more in-depth study is necessary to fully ascertain the tolerance limits toward the presence of humic acid. The Quantifiler trio is reported to detect PCR inhibition at hematin concentrations of about 60 ng/µL or higher. There are no studies on indigo for commercial assays, probably due to the effects of the dye on fluorescence detection.

The SD quants assay [22] showed a similar pattern for hematin, only presenting Cq values outside of the mean ± 0.5 IPC-Cq interval for hematin concentrations ≥ 133.3 µM. On humic acid, NuMY appears to be more sensitive than SD quants, presenting two replicates flagged for inhibition, while SD quants presented none. The SD quants results for indigo replicated the findings for NuMY, by producing too-low IPC-Cq values.

### 3.6. Challenging Biological Samples

The set of challenging samples was divided into three categories differing by biological material and degradation state. The first category consisted of single donor mock samples that were not or were only moderately degraded but represented samples found in typical routine scenarios. The second category included human bone remains of different ages. They served as a model for naturally degraded samples and were used to test the detection of degradation by the assay using the nuRNU/YRS ratio. Finally, the third category included hairs that are of high forensic relevance, as they are very often found at crime scenes [43]. Here, we considered both hair roots and hair shafts. Especially the latter contains only minute amounts of nDNA, which is typically also severely degraded. A summary of the quantification results averaged for the two duplicates is available in Appendix A.

We tested NuMY on seven mock casework samples under controlled casework-type conditions by using GEDNAP proficiency test samples that were DNA-typed previously using STRs and mtDNA (Figure 2). The mtDNA quantification results ranged between 78.9–25,635 mtGE/µL, whereas nDNA quantification yielded lower results, as expected. The nuRNU target fell between 0.004–2.51 ng/µL and YRS results between apparently 0 (or ‘undetermined’ for the four female-source samples) and 3.91 ng/µL. A ratio between nuRNU/YRS quantities was calculated for all male-source samples and most results provided values within our stipulated interval ranging from 0.5 to 2 (Table 2). One sample (GEDNAP_45-S2) yielded a 2.14 nuRNU/YRS ratio, which might represent some DNA degradation that could be caused by repeated freeze–thaw cycles in a similar way to Coriell sample NA18498. Otherwise, the mock casework samples brought ratios within the expected range that were also plausible to the STR genotypes that were established for these samples.

The six DNA extracts from bones were tested with NuMY to gauge performance with naturally degraded DNA. The nuclear targets expectedly gave lower values than the mock casework samples, ranging between 0.44 pg/µL–1.16 ng/µL for nuRNU and 0–0.23 ng/µL for YRS (Figure 2). The observed values for mtDNA ranged between 16.9–660,808 mtGE/µL, which is plausible for such specimens. The 30-year-old bone yielded higher values than the two older bones. Furthermore, the DNA extraction method with improved recovery of shorter DNA fragments (N) yielded nuRNU/YRS ratios indicative of increased DNA fragmentation (Table 2). This finding confirms previous observations by [44] and others for this method.

Due to the different amounts of DNA present in hair roots and shafts, the results are displayed and discussed separately (Figure 2). Hair roots and shafts were collected from one female (female 2) and one male (male 3) individual, while only single hair root samples were available from the other three donors. As expected, DNA quantities for both mtDNA and nDNA targets were higher in root samples than in hair shafts. Nevertheless, nDNA quantification in hair shafts was possible for all three samples, except, as expected, for YRS in the female-source sample. Indeed, all samples of female origin in our study showed no cross-amplification of YRS, indicating that the 473 YRS hits on other chromosomes (see Section 3.1) do not pose a male specificity issue. The male root sample showed on average 4253 times more nDNA than the hair shaft; the same factor in the female hair was only 44. One explanation for these results could be damage resulting from chemical hair treatment, which is more common in women (e.g., bleaching, hair dyes, extreme heat, etc.).

The nuRNU/YRS ratios calculated for all male samples ranged between 0.5–2, except for a hair shaft sample that showed DNA degradation (ratio = 4.55, Table 2). DNA degradation of these samples had been demonstrated for mtDNA targets in a previous study [22]. These results underline the usefulness of such metrics for assessing the quantity, quality, and integrity of samples.

### 3.7. DNA Mixtures

DNA mixtures of female–male components were prepared and quantified using NuMY. While the nuRNU/YRS ratio fluctuated around 0.5–2 for good-quality single-donor samples, the ratios observed for the mixtures (barring those at equal parts) were exceeding the value of 2, as expected for a major female component (Table 2). This quotient increased as the proportion of male DNA in the mixture decreased (Table 2, Figure 1D). However, the minor male component remained detectable in all cases, even at the lowest 1:19 mixture ratio (0.05 ng/µL, Appendix A). For female samples, no YRS signal was obtained, and no template controls typically failed to amplify any of the three quantification targets.

## 4. Conclusions

We present NuMY, a new multiplex qPCR tool intended for the quantification of human nDNA and mtDNA by simultaneously amplifying and detecting autosomal, Y-chromosomal, and mtDNA target sequences. In addition to that, NuMY also features detection of PCR inhibition, and it may be used to assess severe DNA degradation. Thanks to its Y-chromosomal target, NuMY is able to attribute the male sex to unknown single-source stains and quantify the male component of mixed biological traces containing the genetic material of both sexes. NuMY has been developed from SD quants, and it was tested using several sample sets of forensic relevance. The latter included mock casework samples, mixtures, inhibited samples, and naturally degraded samples of varying tissue origins (bone and hair specimens). Moreover, NuMY complies with the established MIQE guidelines and has proven to be a reliable and less expensive alternative to its commercial counterparts.

## Figures and Tables

**Figure 1 genes-14-01645-f001:**
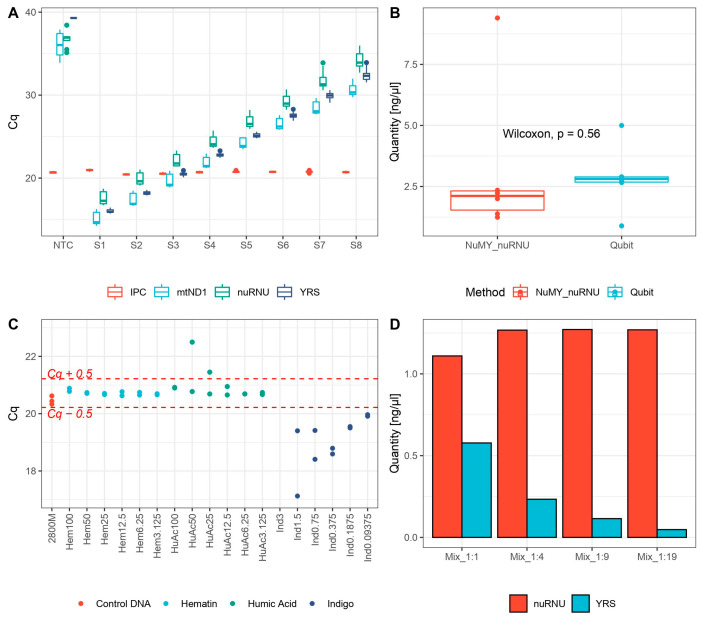
(**A**) C_q_ variability of calibration curve standards (n = 21) and negative controls for all targets: IPC, mtND1, nuRNU, and YRS. While the *C*_q_ values of mtND1, nuRNU, and YRS vary and increase with decreasing template DNA input (Standards 1—62.5 ng/µL and 972,568 mtGE/µL to Standard 8—0.016 ng/µL and 12.5 mtGE/µL), the IPC remains constant around a *C*_q_ value of 21. (**B**) Quantification results of the Coriell samples using the NuMY assay and the Qubit DS HS assay. Mean comparison statistical tests (Wilcoxon non-parametric test) showed *p*-values above 0.05 and thus showed no significant differences between the results obtained by the different methods. (**C**) *C*_q_ values obtained by a control sample (2800 M) and all inhibitor-spiked samples tested using NuMY. Different colors depict distinct inhibitors. Red dashed lines indicate the interval spanning ± 0.5 *C*_q_ values from the mean IPC-*C*_q_ obtained for the 179 calibration curve standard replicates. (**D**) Quantification results of markers nuRNU and YRS obtained from female–male (male being the minor contributor) mixture samples at the following rations: 1:2, 1:5, 1:10, and 1:20.

**Figure 2 genes-14-01645-f002:**
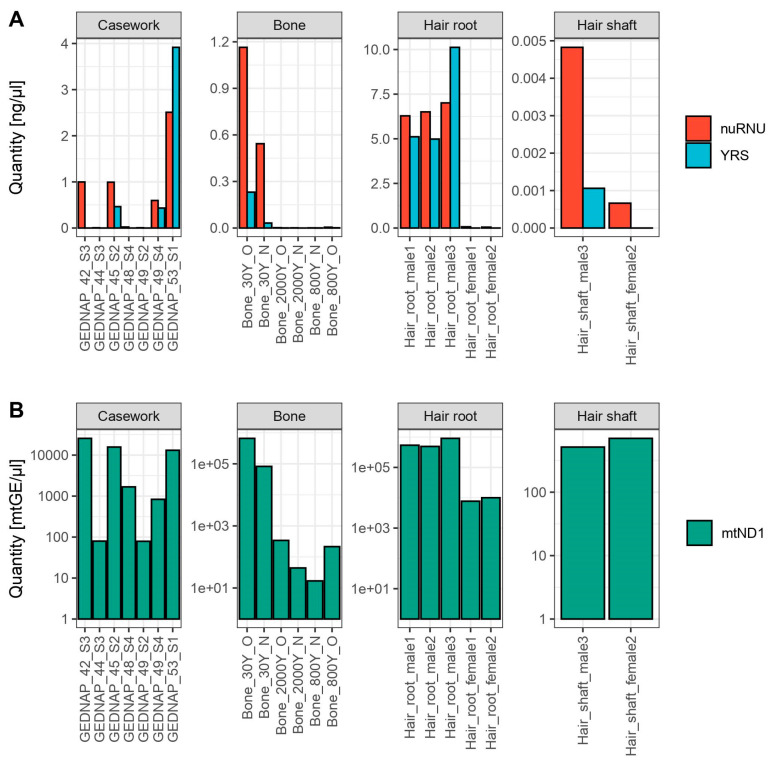
(**A**) Quantification results for markers nuRNU and YRS (ng/µL) for all challenging samples divided according to tissue and degradation state. (**B**) Quantification results for mtND1 (mtGE/µL) for all challenging samples divided according to tissue and degradation state. Mitochondrial DNA results are shown in a logarithmic scale (log_10_) to aid visualization.

**Table 1 genes-14-01645-t001:** NuMY primers and probes and their final concentrations in a 20 µL assay.

Target	Size [bp]	Type	Sequence	Final Concentration
**nuRNU**	70	Forward	GGATTTTTGGAGCAGGGAGA	900 nM
		Reverse	CTGCAATACCAGGTCGATGC	900 nM
		Probe	**FAM** GAGCTTGCTCCGTCCACTCC **BHQ-1**	500 nM
**YRS**	117	Forward	AGTGTTACAGCACTTAAAGGTGT	600 nM
		Reverse	AGGTCTGCAGCTTCATTCT	600 nM
		Probe	**NED** TCTTGCTCACTTCAA **NFQ/MGB**	400 nM
**mtND1**	69	Forward	CCCTAAAACCCGCCACATCT	150 nM
		Reverse	GAGCGATGGTGAGAGCTAAGGT	150 nM
		Probe	**VIC** CCATCACCCTCTACATC **NFQ/MGB**	80 nM
**IPC**	69	Forward	ATCAGCTTAGCGTGCAGTCA	100 nM
		Reverse	TCTTCGTCGTAACGGTGAGC	100 nM
		Probe	**Cy5** GTTGCACTACTTCAGCGTCCCA **BHQ-2**	100 nM
		Template	ATCAGCTTAGCGTGCAGTCAGATAATGTTGCACTAC TTCAGCGTCCCAAGCTCACCGTTACGACGAAGAG	2.5 fM

BHQ-1|2: Black Hole Quencher 1|2, NFQ/MGB: Non-fluorescent Quencher/Minor Groove Binding moiety.

**Table 2 genes-14-01645-t002:** Ratio between the quantification results of nuRNU and YRS for all relevant samples: controls and Coriell samples to establish a baseline interval, challenging samples (mock casework, bones, and hairs), and mixtures. Female samples did not produce YRS results and, consequently, neither nuRNU/YRS ratio is described as such in the table.

Type	Sample	Ratio nuRNU/YRS
Control	2800M	1.82
	007	0.58
Coriell	NA07029	1.92
	NA06994	1.01
	NA07000	female
	NA10540	1.99
	NA11200	0.46
	NA18498	2.3
Mock casework	GEDNAP_42-S3	female
	GEDNAP_44-S3	female
	GEDNAP_45-S2	2.14
	GEDNAP_48-S4	female
	GEDNAP_49-S2	female
	GEDNAP_49-S4	1.38
	GEDNAP_53-S6	0.64
Bone	Bone_30Y-O	5.04
	Bone_30Y-N	16.87
	Bone_2000Y-O	41.81
	Bone_2000Y-N	n.a.
	Bone_800Y-O	5.55
	Bone_800Y-N	7.74
Hair root	Hair_root_male1	1.23
	Hair_root_male2	1.31
	Hair_root_male3	0.69
	Hair_root_female1	female
	Hair_root_female2	female
Hair shaft	Hair_shaft_male3	4.55
	Hair_shaft_female2	female
Mixture	Mix_1:1	1.92
	Mix_1:4	5.43
	Mix_1:9	11.06
	Mix_1:19	26.56

## Data Availability

The data presented in this study are available in the article or Appendix A.

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
