# Peer review of "NuMY—A qPCR Assay Simultaneously Targeting Human Autosomal, Y-Chromosomal, and Mitochondrial DNA"

_genes, 2023, doi:10.3390/genes14081645_

Round 1

Reviewer 1 Report

As a forensic geneticist, I really appreciate the proposed manuscript: having the possibility of quantifying both nuclear DNA (total and Y chromosome) and mitochondrial DNA at the same time can be of valuable help in the management of laboratory activities, especially when the quantity or the quality of DNA is very low.

I have just some suggestions:

- explain why it is not possible with this proposed method to predict also the evaluation of the nuclear and mitochondrial DNA degradation index

- place greater emphasis on the importance and benefits of being able to simultaneously quantify total nuclear DNA, Y chromosome, and mitochondrial DNA in forensic settings

- move to the materials and methods what is reported from lines 61 to 77, avoiding repetitions of the same concepts.

Author Response

We would like to thank both reviewers for their insightful comments, please see below the addressed points. Alterations in the manuscript are highlighted in yellow for convenience and line numbers are stated per point.

Reviewer 1:

As a forensic geneticist, I really appreciate the proposed manuscript: having the possibility of quantifying both nuclear DNA (total and Y chromosome) and mitochondrial DNA at the same time can be of valuable help in the management of laboratory activities, especially when the quantity or the quality of DNA is very low.

I have just some suggestions:

- explain why it is not possible with this proposed method to predict also the evaluation of the nuclear and mitochondrial DNA degradation index

NuMY contains one mtDNA and one Y-chromosomal target instead of two mtDNA targets (as in SDquant), which is why a mtDNA degradation index is not available. However, information about the degradation state of a sample can be inferred through the two nuclear targets (autosomal and Y-chromosome). We have included this information in the Material and Methods section.

- place greater emphasis on the importance and benefits of being able to simultaneously quantify total nuclear DNA, Y chromosome, and mitochondrial DNA in forensic settings

The revised manuscript includes more specific information in the Introduction section.

- move to the materials and methods what is reported from lines 61 to 77, avoiding repetitions of the same concepts.

The concept of this assay is central to the manuscript, which is why we suggest to keep information in Introduction and Material and Methods sections.

Reviewer 2 Report

The study definitely targets an area of forensic DNA typing, that can be improved. However I have some comments:

While targeting mtDNA targets in multiplex qPCR, one should be sure that the quantitation results for mtDNA are not biased by NUMTs. Although NUMTs can involve the entire mtDNA molecule, NUMT breakpoints were more common in the non-coding D-loop and thus ND1 target should not be a problem. However at least in silico search or better an experiment (mock samples) with exonuclease V should provide an answer.

What I also miss is the specie specificity test. The qPCR assay should be tested (in vitro) for a panel of microorganisms and the most common vertebrates.

Author Response

We would like to thank both reviewers for their insightful comments, please see below the addressed points. Alterations in the manuscript are highlighted in yellow for convenience and line numbers are stated per point.

Reviewer 2:

The study definitely targets an area of forensic DNA typing, that can be improved.

However I have some comments:

While targeting mtDNA targets in multiplex qPCR, one should be sure that the quantitation results for mtDNA are not biased by NUMTs. Although NUMTs can involve the entire mtDNA molecule, NUMT breakpoints were more common in the non-coding D-loop and thus ND1 target should not be a problem. However at least in silico search or better an experiment (mock samples) with exonuclease V should provide an answer.

While we agree with the reviewer that NUMTs are generally an issue with mtDNA typing, their impact on the quantitation result is minor in most of the cases, as the majority of NUMTs are single copy. We believe that we assessed this comment by providing the results of an extensive blast search of all primers-probe combinations of this assay (Supplementary Table S4). This search showed that MtND1 probe and primers had only one hit at the desired location in the mtDNA.

What I also miss is the specie specificity test. The qPCR assay should be tested (in vitro) for a panel of microorganisms and the most common vertebrates.

We understand and generally agree with the request of the Reviewer 2. We suggest this to be performed in the course of internal validation. of the assay (which is beyond the scope of this study).

Round 2

Reviewer 2 Report

I would like to thank the authors for addressing my comments. The revised version can be accepted in present form.